# Statistical Study on the Motivation of Patients in the Pediatric Dentistry

**DOI:** 10.3390/children9111782

**Published:** 2022-11-20

**Authors:** Lucian Josan, Sorana Maria Bucur, Mariana Păcurar, Elina Teodorescu, Andreea Sălcudean, Cristina Stanca Molnar Varlam, Alina Ormenișan

**Affiliations:** 1George Emil Palade University of Medicine, Pharmacy, Science, and Technology, 540139 Târgu-Mureș, Romania; 2Department of Dentistry, Faculty of Medicine, Dimitrie Cantemir University, 540545 Târgu-Mureș, Romania; 3Orthodontic Department, Dental Medicine, Carol Davila University of Medicine and Pharmacy, 010221 București, Romania

**Keywords:** children, motivation, orthodontics, pedodontics, oral health

## Abstract

Our statistical study included 344 participants selected from the patients of the Pedodontics–Orthodontics Discipline of the Tîrgu-Mureş University of Medicine and Pharmacy. The patients’ age was between 6 and 18 years, with an average of 13.70 and a standard deviation of 4.62. The study participants were informed and agreed to complete two questionnaires of our conception regarding their health status, oral hygiene, and motivation for pedodontics or orthodontic treatment. The results of the two questionnaires were interpreted according to the gender and age of the patients. Data processing was performed with NCSS/PASS Dawson Edition statistical software, using the CHI2 test, considering a *p* of less than 0.05 as significant for comparative results. Results showed that girls were more motivated than boys in addressing pedodontic services due to dental, periodontal, and articular problems. Children, aged between 11 and 14 years, were less intrinsically motivated to solve oral health problems due to their low frequency. The intrinsic motivation for a more beautiful dentition was very strong, regardless of age and sex. Girls were more intrinsically motivated for orthodontic treatment than boys. There was a linear increase together in the age of those who wanted to improve their smile and facial appearance. Children between 11 and 14 years had the best self-perception of the appearance of their teeth, mouth, smile, and facial harmony. The strongest extrinsic motivation for orthodontic treatment came from parents or another doctor. The most important reason for orthodontics was dental malpositions, the last one was the improvement of masticatory efficiency. The extrinsic motivation from parents for orthodontics decreases linearly with age, along with the increase in motivation from the person with whom the participants relate emotionally and from the group of friends.

## 1. Introduction

Defined in a large sense, motivation is the set of dynamic factors that determine the behavior of an individual [1]. Intrinsic or extrinsic motivation produces different effects on the patient’s psyche, depending on their gender and age, temperament traits, and individual personality [1,2].

In the case of young children, positive motivation based on praise, rewards, and extrinsic factors is more effective [3] because they depend on adults and act mainly according to their wishes and under the conditions imposed by them. The pedodontist or the orthodontist has the task of guiding the small patients towards the appropriate treatment, motivating them to accept it. Praise and encouragement are much more useful than punishment at a young age [1,2,3,4].

In the case of young children, the emotional motivation is stronger than the cognitive one, and the little ones often submit to treatment not to lose the affection and protection of their parents or the doctor’s appreciation [5].

Intrinsic motivation is formed later, depending on self-esteem and individual personality [5,6]. Older children or adolescents are intrinsically or cognitively motivated because they realize the dental problems or the present orthodontic anomaly and want to solve them to improve their health [5,6,7]. Affective motivation intervenes when the adolescent wants to improve his appearance to be more appreciated by his entourage, friends or social group, boyfriend or girlfriend, or when the feeling of rivalry appears [5,6,7,8]. Adolescence is a unique stage of life that includes great emotional changes and intense socialization. Awareness of the appearance of the face, teeth, and smile is acute; the motivation for treatment depends very much on the proposed orthodontic appliance, which must change the patient’s appearance as little as possible. Affective motivation or the desire to have a more pleasant physical appearance for self-esteem often replaces an absent or low cognitive motivation [6,7,8]. The type and intensity of motivation are essential for obtaining therapeutic results. In orthodontics, highly intrinsically motivated patients accept long-term treatments, wear appliances that cause discomfort, and respect the doctor’s instructions and appointments; these are data for the success of the treatments.

On the other hand, the motivation related to dental emergencies such as dental abscesses, gangrene complicated with apical periodontitis, and pulpitis is intrinsic and immediate [8].

American and Northern European pediatric dentists give great importance to the prevention of dental diseases and maintaining a good status of oral hygiene in the case of children and adolescents [9,10,11].

They describe how tooth decay and its complications affect children and young people’s ability to chew food, their sleep, and even their ability to learn and complete schoolwork, having a detrimental effect on their quality of life. In their opinion, the presence of dental diseases also affects the self-esteem of children and adolescents. The intrinsic motivation of children and young people for the prevention and early treatment of these conditions has to be stimulated. 

The role of doctors from related specialties is important in the early detection and referral to our services of children and adolescents with dental, periodontal diseases, and undiagnosed orthodontic anomalies [12]. 

Because of the different effects produced by the different types of motivation, it is necessary to use and exploit them individually depending on the case [5,6,7,8,9].

The study objectives were:1Evaluation of the motivation of minor patients for dental treatment;2Determining the patients’ motivation for orthodontic treatment by evaluating the intrinsic reasons for addressability;3To investigate the extrinsic motivation coming from parents, groups of friends, boyfriend/girlfriend, or another doctor regarding orthodontic treatment.

## 2. Material and Methods

The descriptive study was conducted between May and June 2014 on a group of 344 participants who were selected from the patients of the Pedodontics–Orthodontics Discipline of the Tîrgu-Mureş University of Medicine and Pharmacy. This study was approved by the Ethics Committee of UMFST Tg-Mureș by Decision no. 929 of 26.05.2020.


*The inclusion criteria were:*
-patients with a maximum age of 18;-the signing of the informed consent by one of the parents or by the legal tutor.


The study participants were between the ages of 6 and 18, with an average age of 13.70 and a standard deviation of 4.62. All participants were informed and agreed to complete two questionnaires designed by us regarding their health status, oral hygiene, and motivation for pedodontics or orthodontic treatment. The subjects and their parents were informed by the person who gave them the questionnaires about the purpose of the research, the confidentiality of the data obtained, and the use of the resulting data only for scientific purposes. Instructions for completing the questionnaires were provided.

The questionnaires were administered during the patient’s first visit to our department, before the first visit. All investigated subjects agreed to fill in the questionnaires. Their completion was complete and correct so in the end, we obtained 688 complete questionnaires, whose data we could successfully use for conducting the study.

The obtained data were recorded in databases using the Excel utility with the following variables: the initials of the participant’s name, without any role in data processing, mentioned as identification variables; sex–coded M = masculine, F = feminine.


*The variables of the study were:*
Sex of the patients.Age of patients.


The subjects were distributed according to age in three groups, respecting the theoretical classification of the psychology of ages [13].

-the group of those aged up to 11 years;-the group of those aged between 11 and 14 years;-the group of those over 14 years of age.

Data processing was performed with NCSS/PASS Dawson Edition statistical software, using the CHI^2^ test, considering a *p* of less than 0.05 as significant for comparative results.

The questionnaires: 

The first questionnaire has ten questions; the subjects were instructed to answer with “yes” or “no”. The first five questions refer to the state of health of the teeth, the frequency of pain, and the sensitivity due to dental, gingival, periodontal, and periapical pathology.

The last four questions of the first questionnaire refer to the self-perception of the appearance, alignment of the teeth, the mouth and smile, the symmetry and harmony of the face, and the masticatory efficiency. The last question of the first questionnaire refers to self-esteem concerning dental, oral, and facial appearance.

The second questionnaire refers to the motivation for orthodontic treatment. The first three questions (1.1, 1.2, and 1.3) reflect the patients’ intrinsic motivation for orthodontic treatment. The most frequent reasons why patients turn to the orthodontist were investigated: straighter teeth, a beautiful smile, a more pleasing facial appearance, and better masticatory efficiency.

Through questions 2, 3, 4, and 5 we sought to determine the impact of extrinsic motivation from parents, the group of friends, the person with whom the subjects feel emotionally connected, and another doctor on the addressability of orthodontic treatment.

Questions 5–10 of the first questionnaire and questions 1.1, 1.2, and 1.3 of the second one were designed so that their meanings overlap to a certain extent, to validate the correctness of the answers given by the participants in the study, and to obtain better data correlation and accuracy. The questionnaires were drawn up in the form of tables, to be more easily and correctly completed by the children. (Table 1, Table 2).

Questionnaire I 

Your initials ________________ 

Gender: Female ______Male_____________ 

Age in completed years_________________ 

Instructions: Below there are 10 questions to which please answer by circling the answer that suits you. There is no right or wrong answer. It is important to answer all questions honestly.

Questionnaire II 

Your initials ________________ 

Gender: Female ______Male_____________ 

Age in completed years_________________ 

Instructions: Below there are some questions that please answer by circling the answer that suits you. There is no right or wrong answer. It is important to answer honestly to get an accurate assessment of the reason you are visiting the orthodontist.

## 3. Results

Of the subjects investigated, 192 were girls, and 152 were boys. The girls, therefore, constituted 55.81% and the boys constituted 44.19% of the group.

According to age, the three groups were:-the group of those aged up to 11 years included 83 subjects representing 24.12% of the study participants;-the group of those aged between 11 and 14 had 89 subjects, representing 25.87% of the study participants;-the group of those over 14 years of age was 172 subjects, who represent 50% of the study participants.

The statistical interpretation of the results of Questionnaire I according to the gender of the participants is presented in Table 3.

Percentage interpretation of the results for questions 1–5, and 6–10 depending on the gender of the participants are illustrated in Figure 1 and Figure 2.

The statistical interpretation of the results of Questionnaire I according to the age of the patients is shown in Table 4.

Percentage interpretation of the results for questions 1–5, respectively 6–10 depending on the age groups of the participants are illustrated in Figure 3 and Figure 4.

The statistical interpretation of the results of Questionnaire II according to the gender of the participants is presented in Table 5.

The results do not show there is a statistically significant difference between what girls and boys want; 97.91% of girls and 92.11% of boys answered question 1.1 in the affirmative. 

The majority of girls, with a statistically significant difference, 64.58%, want a better masticatory efficiency, while in the case of boys this percentage is 44.74%. 

There is a statistically significant association between gender and the desire to improve their facial appearance and smile, with girls being more strongly motivated, with a percentage of 91.67% compared to 76.31% of boys.

Regarding question 2, 75% of the girls and 76.31% of the boys answered affirmatively, which reflected extrinsic motivation from the parents. 

There is no statistically significant association between the sex of the interviewees and the extrinsic motivation coming from the boyfriend or girlfriend, the percentages of affirmative answers being 31.25% in the case of girls and 28.95% in the case of boys. 

In addition, 31.25% of girls and 36.84% of boys stated that friends motivated them for orthodontic treatment.

Results also showed that 72.92% of girls and 63.16% of boys received the recommendation to consult an orthodontist from another doctor.

The statistical interpretation of the results of Questionnaire I according to the age of the patients is shown in Table 6.

In all three age groups, the intrinsic motivation for adjusting the position of the teeth and a more aesthetic aspect of the dentition is very strong (100%, 95.45%, 93.02%). 

Results showed that 57.14% of the small group, 54.54% of the second age group, and 55.81% of the adolescent group want better masticatory efficiency. 

In the small group, the percentage of those who want to improve their smile and facial appearance is 71.42%, in the 11–14-year-old group it is 90.91%, and in the adolescent group 88.37%. The differences are statistically significant.

The extrinsic motivation for orthodontic treatment coming from parents decreases linearly with increasing age; 85.71% of children up to 11 years, 72.72% of the middle-aged group, and 69% of teenagers answered affirmatively. 

There is a linear increase in the age of those extrinsically motivated by the boyfriend or girlfriend (14.29%, 27.27%, 39.53%). The differences are statistically significant.

The association between age and the affirmative answer to the question that quantifies the extrinsic motivation coming from friends is statistically significant, with 28.57% in the group up to 11 years, 27.24% in those between 11 and 14 years old, and 44.19% in teenagers. 

Results showed that 61.90% of those aged up to 11 years, 72.27% of the 11–14-year-old group, and 67.44% of the over 14-year-old group stated that another doctor recommended them to consult an orthodontist.

## 4. Discussions

### 4.1. Regarding Questionnaire I

The association between gender and the answer to the question “Do your teeth look healthy and clean?” can demonstrate a self-critical spirit and higher claims of girls compared to boys regarding dental health.

Correctly assimilated oral hygiene skills at the age of 11–14, as opposed to a younger age category and young permanent teeth not yet attacked by cariogenic agents for long periods, as in the case of the most “elderly” group, explain the better results of this age group to this question.

Another study by Kumar et al. [14] on 831 children 12 years old in India found the prevalence and severity of dental caries lower among urban children and girls. Oral health knowledge and practices were lower in rural and male children than in urban children and girls. Caries’ experience was significantly associated with gender.

Another study conducted in India on a number of 524 children between the ages of 5 and 13 [15] has found similar results to our study. The lower prevalence of dental caries was in the group of children aged 11–13 years old. Contrary to our study, no statistically significant correlation was found between the prevalence of dental caries and the gender of the participants

The answers to the question “Have you ever felt that your teeth hurt?” correlate positively with those to the question related to the perception of health and dental hygiene; girls are the ones who declare in a much lower percentage than boys that their dental health is good, so it is perfectly normal that pains of dental etiology are more frequent in the case of the female sex. The correlation of the percentage difference between the affirmative answers given by the two sexes to questions 1 and 2 of Questionnaire I shows us that girls are indeed more critical in the self-assessment of the state of dental health. A large study conducted in the United States [16] also found a high prevalence of pain of dental etiology that is not directly correlated with the frequency of carious processes among older children.

A group of American researchers [17] examined the correlation between school absenteeism, children’s academic performance, and dental pathology in a group of 2183 students. Children with dental problems were three times more likely to miss school due to pain than those without dental problems and 2.5 times more likely than those with asthma. Similar to our study, children with poor oral health presented more frequently with dental pain as a symptom. Contrary to our study, the male sex was more affected than the female sex, and adolescents had dental pain as a symptom more frequently than children.

As the American Dental Association [18] claims, in good health and dental hygiene, dental sensitivity decreases or is absent. It explains why boys and the middle-aged group, with a lower frequency of oral and dental pathology, present dental sensitivity in a lower percentage.

Other authors [14,19] note the higher frequency of gingival diseases in the case of female patients due to the hormonal changes that occur during puberty and the alteration of the superficial periodontal tissues so that the microbial dental plaque causes the exaggerated response of these structures.

Our results agree with those of Shishniashvili et al. in a study conducted in Tbilisi [20] on a group of 618 school children aged between 9 and 15. Superficial gingivitis and periodontitis were more frequent and had more severe forms in the case of girls aged 12–13 years. The poor state of oral hygiene and orthodontic anomalies were favorable factors for these pathologies.

Correlating the answers to questions 1 and 5, we notice that the frequency of dental and periodontal abscesses is higher in the case of girls, caused by their poorer health condition and dental hygiene. Those with a higher frequency of periodontal or periapical abscesses are children up to 11 years old. Similar to our study, Azodo et al. found a significant frequency of dental abscesses among children aged between 6 and 11—40.5% of all children with this pathology. Contrary to our study, the authors found most cases in boys, with a percentage of 64.3% of the total of investigated subjects [21].

Dentomaxillary anomalies are impossible to diagnose correctly by patients, who nevertheless notice the imperfections related to the position of their teeth, which they want to remedy. Although 2/3 of both sexes notice dental malpositions, the face physiognomy, the appearance of the lips, and the smile give the majority of the study subjects a positive self-image perception.

The results for the question “Do your teeth look straight?” demonstrates the impact on the appearance of the dentition during the period of dental change in the case of children up to 11 years old and the high demands of adolescents regarding dental esthetics, as they may be shy and inhibited due to dental malpositions that affect their self-esteem and quality of life [22]. The best results were obtained by the second age group.

Dutra M.B. et al. [23] aimed to evaluate the perception of orthodontists, general dentists, and ordinary people regarding smile esthetics. The most aesthetic, pleasant, and attractive female smile was the one with the upper lip over the gingival margin of the upper incisors, exposing the entire length of the crowns. In the case of the male gender, the most pleasant smile perceived by ordinary people was that with the edge of the upper lip placed over the gingival edge of the crowns of the upper incisors. Dentists and orthodontists considered the most aesthetic situations to cover the incisors with 2 mm by the upper lip and the overlap of the lower limit of the upper lip over the gingival margin.

Another study conducted by Farzanegan et al. in Iran finds that orthodontists and general dentists equally appreciate each aesthetic component of the smile (teeth and soft tissues), while ordinary people do not perceive the effects of the anatomic details that make a pleasant smile [24]. It explains our high percentage of patients satisfied with the appearance of their mouths and smiles. The orthodontists perceive finer facial asymmetries than ordinary people, which also explains our high percentage of patients satisfied with the harmony and symmetry of their faces [25].

Adolescents are more self-critical and more interested in their facial appearance compared to the other two age groups. Adolescents’ self-esteem suffers due to orthodontic anomalies that affect facial harmony. A study in Saudi Arabia [26] found similar results; adolescents were the most self-critical regarding facial harmony. 

Arqoub et al. [27] found the two sexes appreciate facial harmony in similar percentages. A study conducted in Brazil [28] by Faverani et al. concluded, contrary to our results, that the female gender is more self-critical regarding facial symmetry and harmony.

We found girls had a much lower percentage than boys of appropriate dental health and hygiene; a significantly higher percentage of dental pathology was present, therefore the number of masticatory units may be lower than normal. More frequent tooth pain and sensitivity in the case of girls (see the answers to questions 2 and 3) may determine the avoidance of using masticatory groups; the consequences are occlusal, muscular, joint imbalances, and low masticatory efficiency.

Temporomandibular dysfunctions are more frequent in the case of females due to more pronounced ligamentous laxity; pain and limitations at these joints are reasons for low masticatory efficiency. The data presented by Poveda and Graf [29,30] agree with those we found. They claim that age is a risk factor for TMJ pathology in the case of adolescents and young adults; the symptoms of TMJ dysfunction are serious, and the treatments are complex. The stress associated with bruxism and parafunction causes pain in the facial muscles. Orthodontic treatment is not a cause of this pathology [29].

Most of our patients, regardless of gender, have a positive perception of their facial appearance and smile. The lower percentages of affirmative answers compared to those given by the respondents to questions 7 and 8 show that the study participants were influenced by the perception of dental anomalies shown by question 6, a perception that is investigated in the case of question 10 by using the word “speech”. During the speech, dental malpositions can be noticed by those we address [31].

The 11- to 14-year-old group reported the best results regarding oral health status, self-perception of the appearance of the mouth, and smile; that is why they feel comfortable when smiling or talking to others. Another explanation would be the indifference or neglect towards physical appearance characteristic of puberty.

### 4.2. Regarding Questionnaire II

The results of question 1.1 can be correlated with the participants’ answers to question 6, “Do your teeth look straight?” from the first questionnaire, which demonstrates dissatisfaction with dental aesthetics; the desire to improve it and thereby improve the self-image can explain the overwhelming number of those who, regardless of gender and age, want more beautiful and straighter teeth.

The motivation of the female sex to improve occlusal relations and masticatory efficiency is much stronger than that of the male sex. These results correlate positively with those given by the participants to question 9 of Questionnaire I, where a percentage of 58.33% of the girls were satisfied with their masticatory efficiency, compared to 76% of the boys. Results obtained according to the age of the subjects for the question “Do you want to be able to chew better?” correlates with those obtained for question 9 of the first questionnaire, “Can you chew food well?”.

The higher percentage of boys satisfied with the appearance of their mouth and smile justifies the answer to question 1.3 of Questionnaire II, in which girls seem to have a stronger objective motivation for changing their smile and facial appearance. The motivation can be subjective, with girls being more demanding regarding the aesthetic aspect than boys. The desire to look better, to bring improvements to the physiognomy, and smile motivates patients to start any orthodontic and/or dentofacial orthopedic treatment. The interest in external appearance increases with age, along with self-esteem and the desire to integrate socially and into a group of friends.

A study from Great Britain [32] found that when ANB angle values from the Tweed analysis varied by more than 5 degrees from the normal ones, the face was considered less attractive. Increased values of the anterior face height were considered unattractive in the case of women and on the contrary attractive in the case of men.

The extrinsic motivation for orthodontic treatment with parents as initiators is very strong in the case of both sexes [33], the very close percentages being natural because parents’ care for their children does not depend on their gender. The extrinsic motivation from the parents decreases linearly with the increase in age of the participants, and the influence of the parents’ advice decreases with the growth and maturation of the children.

There is a linear tendency to increase with age in extrinsic motivation for orthodontic treatment from the opposite sex since sexual and affective maturation causes young people to be receptive to the wishes of the person of the opposite sex they like. The desire to please is an emotional motivation for the recovery of self-image in adolescents.

At young ages, aesthetic requirements are modest; they are not a criterion in establishing friendships between children, so the extrinsic motivation for orthodontic treatment coming from friends has low values. A significant increase with age was observed in the percentage of those influenced by the social group they belong to; the highest percentage is in teenagers, who are more receptive to the advice of friends and eager to improve their self-image for social integration.

Williams et al. [34] demonstrated using the questionnaire method that the most important motivation of the participants for orthodontic treatment was, similar to our study, the solution of dental malpositions (80%); 69% of the participants wanted orthodontics for the prevention of dental and periodontal problems and 68% aimed to improve their self-esteem. The teenage girls were motivated by obtaining a more beautiful smile and improving their self-image. The teenagers wanted to improve their social life.

In another retrospective study conducted in Finland [35], the patients’ motivation for orthodontic treatment was investigated through a questionnaire of 14 items. The most important reason for addressability was problems related to masticatory efficiency (68%). Another reason (36%) was the improvement of facial aesthetics. In addition, 32% of the patients wanted to resolve temporomandibular joint dysfunctions accompanied by pain. Similar to our study, girls were more dissatisfied with facial appearance than boys, but the authors did not find a statistically significant association between gender and this parameter; 42.10% of girls and 22.22% of boys wanted a more pleasing facial appearance.

A study conducted in Germany by Reinhardt et al. [36] found that less than 10% of the patients were very well intrinsically motivated, 69 less motivated, and 179 without direct motivation. The last result shows that the motivation for orthodontics comes from other sources—extrinsic or indirect motivation. The very well intrinsically motivated patients could wear orthodontic appliances that cause discomfort and accept long-term treatment, contrary to the other two groups. The results of the treatments were influenced by the type and intensity of the initial motivation of the patients.

A study conducted in London [37] investigated the motivation and expectations of minor patients and their parents, demonstrating the effectiveness of the questionnaire method in finding the motivational factors of addressability to the orthodontic service. Through the interview method, 32 reasons for the children’s decision for orthodontics and 35 reasons why the parents brought them to the orthodontist were found. Based on their answers, two questionnaires were created: one addressed to patients and the other for parents. The results showed that adolescents were interested in improvements in facial and dental aesthetics, while their parents were concerned with the prevention of future problems for their children. The study demonstrated the effectiveness of the questionnaire method in dental research, in sectors where subjective opinions have a significant influence.

Hamdan [38] investigated and compared the real need for orthodontic treatment with the subjective need for treatment. The Index of Orthodontic Treatment Need (IOTN) was used to determine the real need for orthodontic treatment. The results showed that 93% of the study subjects wanted the treatment for esthetic reasons, and in 43% of them, the physiognomic disorders were noticed by parents or friends. In 43% of the cases, the patient wanted the treatment; in 30% of the cases, the parents initiated the treatment; in 18%, the general practitioner; and in 9%, the pedodontist. The study demonstrated, similar to our study, that the perception of the need for treatment depends on several factors, and the motivation for orthodontic treatment comes from several sources. The orthodontist perceived the need for treatment more accurately than the patients, although the patients were aware of the existence of a physiognomic disorder.

Wedrychowska and Syrynska [33] find that the desire to improve dental aesthetics was the main motivation for the addressability of minor patients in Orthodontics. As the age of the subjects increases, the extrinsic motivation for treatment decreases. Less than 5% of minor patients went to the orthodontist because other children laughed at their facial appearance. Between 63–67% of parents stated that they pressure their children to request orthodontic treatment in order to avoid future complaints from them. No statistically significant correlations were found between gender and motivation for treatment.

Daniels et al. [39] found contrary results to our study; parents reported a stronger motivation for orthodontic treatment of their adolescent children than they did, a result that was not significantly correlated with patient cooperation. The intrinsic motivation for the treatment was the one that determined compliance with the recommendations given during the therapeutic sessions. A similar result was obtained by Hiemstra et al. in Holland [40].

Dias and Gleiser [41] describe the real need for orthodontic therapy as greater than that considered by study participants, parents, and children. Parents were more motivated than children to start orthodontic treatment, a result similar to our study for the young age group, up to 11 years. Girls were much more interested in undergoing treatment for aesthetic improvements than boys.

## 5. Conclusions

Children between 11 and 14 years have the best self-perception of the appearance of teeth, mouth, smile, facial harmony, symmetry, and the highest self-esteem and social comfort caused by these aspects.Older children, aged between 11 and 14 years, are less intrinsically motivated to solve oral health problems due to the lower frequency of dental, periodontal, and articular pathology.Girls are more motivated than boys in addressing pedodontic services due to dental, periodontal, and articular problems. Girls are also more intrinsically motivated for orthodontic treatment than boys.The intrinsic motivation for a more beautiful dentition and a more harmonious arrangement of the teeth is very strong, regardless of age and sex.The strongest extrinsic motivation for orthodontic treatment comes from parents or another doctor with a specialty related to orthodontics.There is a linear increase together in the age of those who want to improve their smile and facial appearance.Within intrinsic motivation, the most important issue is dental malpositions, and the last one is the improvement of masticatory efficiency.The extrinsic motivation for orthodontic treatment coming from parents decreases linearly with age.The extrinsic motivation for orthodontic treatment coming from the person with whom the participants relate emotionally and from the group of friends increases linearly with age.

## Figures and Tables

**Figure 1 children-09-01782-f001:**
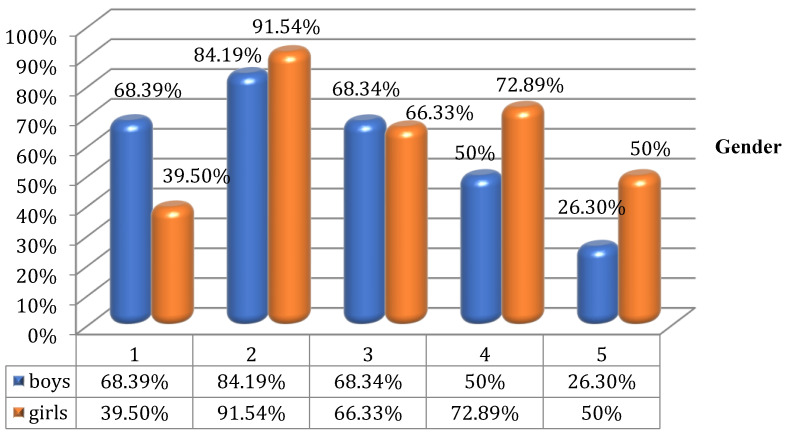
Affirmative answers to questions 1–5 of Questionnaire I according to gender.

**Figure 2 children-09-01782-f002:**
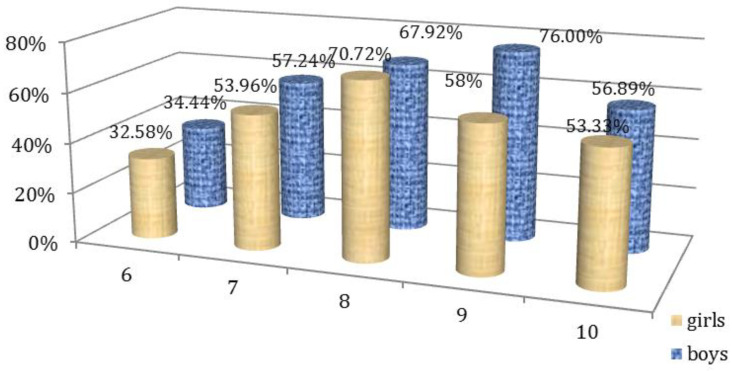
Affirmative answers to questions 6–10 of Questionnaire I according to gender.

**Figure 3 children-09-01782-f003:**
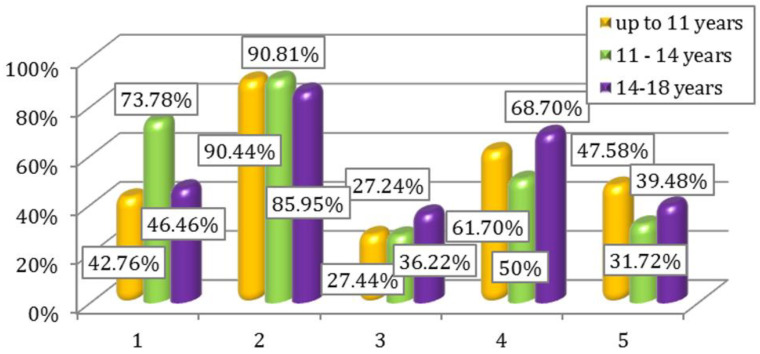
Affirmative answers to questions 1–5 of Questionnaire I according to age.

**Figure 4 children-09-01782-f004:**
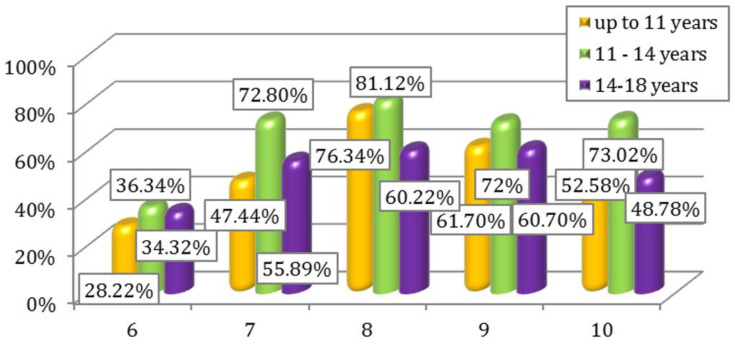
Affirmative answers to questions 6–10 of Questionnaire I according to age.

**Table 1 children-09-01782-t001:** Questionnaire I.

1. Do your teeth look healthy and clean?	Yes	No
2. Have you ever felt that your teeth hurt?	Yes	No
3. Can you eat/drink cold, warm, or sweet, sour foods/liquids?	Yes	No
4. Have you ever had red, swollen, painful, or bleeding gums?	Yes	No
5. Have you ever had an abscess (pus) in a tooth/molar?	Yes	No
6. Do your teeth look straight?	Yes	No
7. Do you think your mouth and smile are beautiful?	Yes	No
8. Do you think your face is symmetrical and harmonious?	Yes	No
9. Can you chew food well?	Yes	No
10. Do you feel comfortable smiling/talking to others?	Yes	No

**Table 2 children-09-01782-t002:** Questionnaire II.

1. You went to the orthodontist because:		
s1.1 Do you wish to have more beautiful and straighter teeth?	Yes	No
1.2 Do you want to be able to chew better?	Yes	No
1.3 Do you want to improve your smile or facial appearance?	Yes	No
2. Did your parents tell you that your teeth should be more straight?	Yes	No
3. Did your girlfriend/boyfriend tell you that you would be prettier with straighter teeth?	Yes	No
4. Have your friends told you that you should straighten your teeth?	Yes	No
5. Has another doctor recommended you to see an orthodontist?	Yes	No

**Table 3 children-09-01782-t003:** The results of Questionnaire I according to the gender of the subjects.

	Chi 2	*p*-Value
Patient gender versus question 1	14.143	0.00016
Patient gender versus question 2	2.297	0.128
Patient gender versus question 3	0.059	0.805
Patient gender versus question 4	9.539	0.002
Patient gender versus question 5	9.951	0.002
Patient gender versus question 6	0.016	0.903
Patient gender versus question 7	1.411	0.233
Patient gender versus question 8	0.118	0.731
Patient gender versus question 9	2.976	0.082
Patient gender versus question 10	0.238	0.623

**Table 4 children-09-01782-t004:** The results of Questionnaire I according to the age of the subjects.

	Chi 2	*p*-Value
Patient age versus question 1	10.024	0.006
Patient age versus question 2	0.918	0.624
Patient age versus question 3	1.718	0.432
Patient age versus question 4	4.879	0.858
Patient age versus question 5	2.233	0.332
Patient age versus question 6	0.667	0.712
Patient age versus question 7	5.962	0.054
Patient age versus question 8	7.368	0.022
Patient age versus question 9	1.987	0,358
Patient age versus question 10	6.992	0.036

**Table 5 children-09-01782-t005:** The results of Questionnaire II according to the gender of the subjects.

	Chi 2	*p*-Value
Patient gender versus question 1.1	3.234	0.066
Patient gender versus question 1.2	6.764	0.008
Patient gender versus question 1.3	7.790	0.006
Patient gender versus question 2	0.048	0.432
Patient gender versus question 3	0.117	0.735
Patient gender versus question 4	0.586	0.431
Patient gender versus question 5	1.875	0.175

**Table 6 children-09-01782-t006:** The results of Questionnaire II according to the age of the subjects.

	Chi 2	*p*-Value
Patient age versus question 1.1	3.087	0.202
Patient age versus question 1.2	0.049	0.961
Patient age versus question 1.3	7.978	0.017
Patient age versus question 2	3.970	0.132
Patient age versus question 3	8.767	0.011
Patient age versus question 4	9.458	0.008
Patient age versus question 5	2.464	0.282

## Data Availability

Data supporting reported results can be found by contacting Sorana Maria Bucur at bucursoranamaria@gmail.com.

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
