# Peer review of "Statistical Study on the Motivation of Patients in the Pediatric Dentistry"

_children, 2022, doi:10.3390/children9111782_

Round 1
Reviewer 1 Report
I have the following concerns:
1. How they conducted the questionnaires to the below 10 years old children?
2. How did they certify that the response was their real perception? (Normally there's a literacy test to ensure the participants have the required level of understanding);
Author Response
- How they conducted the questionnaires to the below 10 years old children?
The questionnaire was addressed individually in oral form and completed/filled out by the doctor.
The items were explained in simple words, according to the child's level of understanding. If there were any misunderstandings or clarification needed, that was addressed too.
2. How did they certify that the response was their real perception? (Normally there's a literacy test to ensure the participants have the required level of understanding);
The study was conducted on normally developed children, without any intellectual impairment or verbal language problems.
All the children included in the sample were integrated into normal schools, where there is a formal assessment for school readiness and there are no children with mental deficiency or literacy problems.
The children's level of their understanding was checked empirically by an initial informal discussion with each child about their name, age, school interests, and preferred activities. The discussion is also meant to reduce anxiety about dental treatment in children and raise treatment compliance.
Reviewer 2 Report
Dear authors,
congratulations for the interesting topic addressed; patient motivation and compliance remain key elements in pedodontics and orthodontics.
To facilitate the presentation of the results, please specify the following:
-did the patients present themselves for consultation on their own initiative or on the recommendation of other doctors?
-was the questionnaire formulated together with a psychologist?
-has the questionnaire been validated?
Author Response
Thank you very much for the time spent reading and evaluating our article. The following are our responses to your concerns.
1. Was the questionnaire formulated together with a psychologist?
The questionnaire was not formulated by a psychologist, but the doctor that conceived it has around 30 years of experience in working with children in orthodontic care and a genuine interest in psychological aspects of dental treatment, confirmed by the CV research publications.
2. Has the questionnaire been validated?
This is a pilot study but validation has to be considered in the future, starting from the results obtained.
An article similar to ours:
Becker A, Shapira J, Chaushu S. Orthodontic treatment for disabled children: motivation, expectation, and satisfaction. Eur J Orthod. 2000 Apr;22(2):151-8. doi: 10.1093/ejo/22.2.151. PMID: 10822888.
Reviewer 3 Report
Statement on page 13 point 4 (row 546) was repeated at point 6 (row 552).
Please delete point 6
Author Response
Statement on page 13 point 4 (row 546) was repeated at point 6 (row 552).
Please delete point 6
Thank you, we have deleted point 6.
Thank you very much for the time spent carefully reading and evaluating our article.